# Recognition and Chaperoning by Pex19, Followed by Trafficking and Membrane Insertion of the Peroxisome Proliferation Protein, Pex11

**DOI:** 10.3390/cells11010157

**Published:** 2022-01-04

**Authors:** Katarzyna M. Zientara-Rytter, Shanmuga S. Mahalingam, Jean-Claude Farré, Krypton Carolino, Suresh Subramani

**Affiliations:** Section of Molecular Biology, Division of Biological Sciences, University of California, San Diego, CA 92093-0322, USA; kzientararytter@ucsd.edu (K.M.Z.-R.); sxm1505@case.edu (S.S.M.); jfarre@mail.ucsd.edu (J.-C.F.); krypton.carolino@gmail.com (K.C.)

**Keywords:** peroxisomal membrane protein, peroxisome proliferation protein, peroxisome division, Pex11, Pex19

## Abstract

Pex11, an abundant peroxisomal membrane protein (PMP), is required for division of peroxisomes and is robustly imported to peroxisomal membranes. We present a comprehensive analysis of how the *Pichia pastoris* Pex11 is recognized and chaperoned by Pex19, targeted to peroxisome membranes and inserted therein. We demonstrate that Pex11 contains one Pex19-binding site (Pex19-BS) that is required for Pex11 insertion into peroxisomal membranes by Pex19, but is non-essential for peroxisomal trafficking. We provide extensive mutational analyses regarding the recognition of Pex19-BS in Pex11 by Pex19. Pex11 also has a second, Pex19-independent membrane peroxisome-targeting signal (mPTS) that is preserved among Pex11-family proteins and anchors the human HsPex11γ to the outer leaflet of the peroxisomal membrane. Thus, unlike most PMPs, Pex11 can use two mechanisms of transport to peroxisomes, where only one of them depends on its direct interaction with Pex19, but the other does not. However, Pex19 is necessary for membrane insertion of Pex11. We show that Pex11 can self-interact, using both homo- and/or heterotypic interactions involving its N-terminal helical domains. We demonstrate that Pex19 acts as a chaperone by interacting with the Pex19-BS in Pex11, thereby protecting Pex11 from spontaneous oligomerization that would otherwise cause its aggregation and subsequent degradation.

## 1. Introduction

Peroxisomes are involved in long-chain fatty acid oxidation and ROS balance [1]. Due to their variable function, they are versatile organelles whose size, shape, and number, as well as content, adapt to environmental requirements [2,3]. Peroxisomes are maintained by proliferation of pre-existing peroxisomes or by de novo synthesis from the endoplasmic reticulum (ER) [4,5]. Both pathways contribute to the cellular pool of peroxisomes and require the import of peroxisomal membrane proteins (PMPs) [6,7,8]. Peroxisomal division that occurs through growth and division of pre-existing peroxisomes requires, besides the membrane fission machinery whose components are shared between mitochondria and peroxisomes [9,10,11,12], the PMP, Pex11, which remodels peroxisomal membranes prior to division. Pex11, one of the most abundant peroxins in the peroxisomal membrane [13], is well conserved in other eukaryotes (Figure 1). Depending on the species, from three to five Pex11 proteins have been identified. In *Arabidopsis thaliana*, five homologs of yeast Pex11 are known, which fall into two clades [14,15]. In mammals, three isoforms of Pex11 have been characterized: Pex11α, Pex11β, and Pex11γ [15,16,17,18,19,20,21], where Pex11α and Pex11β share domain and motif architecture [16], and Pex11γ is the most distinct member [22].

Despite the crucial role of Pex11 as a player in peroxisome division, our knowledge about Pex11 import, in comparison with the import of peroxisomal matrix proteins, remains rather limited. Pex11 is a Class 1 PMP, whose transport to peroxisomes is proposed to be mediated by the import receptor, Pex19, which binds to Pex19-binding sites (Pex19-BSs) present in PMPs [23,24,25]. Consequently, cells with a deficiency of Pex19 lack peroxisomal membrane structures [26,27,28,29]. Pex19 serves not only as the PMP import receptor, but also as a chaperone that interacts with their membrane peroxisome-targeting signals (designated mPTSs) that are often located within hydrophobic regions [23,30]. Unfortunately, the mPTSs of only a few integral PMPs have been defined, without revealing any strictly-conserved mPTS consensus. Interestingly, in the yeast *Pichia pastoris* (now renamed *Komagataella phaffi)*, Pex19 seems not to function solely as a PMP protein import receptor, but in cooperation with the PMP Pex3, facilitates the insertion and orientation of PMPs in the peroxisomal bilayer [27,30].

Importantly, because of the pleiotropic effect of loss of Pex19 on most or all PMPs, it has been difficult to dissect its transport and chaperone roles, from its role in PMP insertion at the peroxisome membrane. To shed some light on these aspects, we investigated how the *P. pastoris* Pex11 (PpPex11) is recognized and chaperoned by Pex19, followed by its targeting and insertion into peroxisome membranes. We reveal that, contrary to other PMPs for which multiple Pex19-BSs were determined [31,32,33,34], Pex11 contains only one detectable Pex19-BS that also serves as an mPTS, which is not in the vicinity of any predicted hydrophobic domains. However, Pex11 also contains a separate, Pex19-independent mPTS that is sufficient to traffic Pex11 to peroxisomes, but is insufficient for insertion of Pex11 into the peroxisome membrane. We show that although Pex19 plays a specific and essential role in Pex11 stability and its proper insertion into the peroxisomal membrane, it is not solely responsible for Pex11 transport to peroxisomes. Thus, Pex11 trafficking to the peroxisome membrane could use non-canonical mechanisms, independent of direct Pex19 binding. 

## 2. Materials and Methods

### 2.1. Molecular Biology Techniques for Plasmid Construction

Plasmids used in this work are listed in Appendix A. For site-directed mutagenesis, the QuikChange II Site-Directed Mutagenesis Kit (Agilent Technologies, Santa Clara, CA, USA, Cat#200523) was used. All plasmids were checked by restriction digestion and/or by DNA sequencing. Conventional techniques were used for *Escherichia coli* transformation.

### 2.2. Yeast Cells, Transformation, and Growth Conditions

The *P. pastoris* strains used in this work are in Appendix A. Media used to grow strains include: YPD (1% wt:vol yeast extract, 2% wt:vol peptone, and 2% wt:vol glucose) and methanol medium (1.7 g/L yeast nitrogen base without amino acids and ammonium sulfate, 0.05% wt:vol yeast extract, 0.5% wt:vol ammonium sulfate, 0.5% vol:vol methanol). Histidine (50 mg/L) and/or arginine (50 mg/L) were added when needed. All cultures were grown at 30 °C. YNB solution (1.7 g/L yeast nitrogen base without amino acids and ammonium sulfate) was used to wash cells. Cells were transformed by electroporation, as described previously [35].

### 2.3. Protein Expression and Purification

GST-Pex19 was purified from *E. coli* BL21 (DE3) cells (Novagen, Madison, WI, USA, Cat# 69450) by GST agarose affinity chromatography. Cells were diluted from an overnight culture and diluted to reach 0.4–0.6 OD600 and induced with 1 mM IPTG (GoldBio, St Louis, MO, USA, Cat# 12481C25) for 3 h at 37 °C. After induction, cells were collected, resuspended in ice-cold lysis buffer (50 mM Tris, pH 8.0 containing 50 mM NaCl, 5 mM EDTA, 1 µg/mL leupeptin, 0.15 mM PMSF, 1 mM DTT, 5 mg lysozyme (Sigma-Aldrich, St. Louis, MO, USA, Cat#L6876) and 1× protease inhibitor cocktail (Sigma-Aldrich, Cat#P8465)), disrupted by sonication (Branson, Danbury, CT, USA, Sonifier 250; 40% amplitude, 10 s sonication and 30 s on ice, total of three cycles) and lysate was spun at 48,000 g (LS55 ultracentrifuge, SW41 Ti rotor, Beckman Coulter, Brea, CA, USA) for 20 min at 4 °C to obtain a soluble fraction. The soluble fraction was incubated with Glutathione Agarose 4B (Prometheus, San Diego, CA, USA, Cat#20-452) for 1 h at room temperature and spun at 500 g for 5 min to remove unbound proteins. Later, the beads were washed with DPBS and the bound protein was eluted using 20 mM reduced L-glutathione (Sigma-Aldrich, Cat#G4251) in 50 mM Tris pH 8.0. Purified recombinant protein was quantified by Coomassie Brilliant Blue staining. 

For protein in vitro binding assay, plasmids containing sequences of His_6_-tagged GFP-Pex11s (Appendix A) were expressed in *E. coli* OverExpress™ C43(DE3) cells (Sigma-Aldrich, Cat# CMC0019) recommended for effective expression of membrane proteins with increased solubility. Cells were diluted from an overnight culture and diluted to reach 0.4–0.6 OD600 and induced with 1 mM IPTG for 6 h at 37 °C. After induction, cells were collected, resuspended in ice-cold lysis buffer (50 mM Tris, pH 8.0 containing 50 mM NaCl, 1 µg/mL leupeptin, 0.15 mM PMSF, 1 mM DTT, 5 mg lysozyme and 1× protease inhibitor cocktail), disrupted by sonication (Branson Sonifier 250; 40% amplitude, 10 s sonication and 30 s on ice, total of five cycles) and centrifuged (LS55 ultracentrifuge, SW41 Ti rotor) at 48,000× *g* for 20 min at 4 °C to obtain the soluble fraction. 

### 2.4. Protein In Vitro Binding Assay 

For the Pex19-Pex11 in vitro binding assay, 500 μL of the soluble fraction of protein lysate of certain His_6_-tagged GFP-Pex11s was mixed with 200 μL of the soluble fraction of protein lysate of GST-Pex19 and 300 μL of Binding buffer (50 mM Tris, pH 8.0, 50 mM NaCl), incubated with Glutathione Agarose 4B (Prometheus, Cat#20-452) for o/n at 4 °C and spun at 1000× *g* for 1 min to remove unbound proteins. Then, the beads were washed five times with ice-cold Binding buffer (50 mM Tris, pH 8.0 containing 50 mM NaCl, 5 mM EDTA). Proteins were eluted with 50 µL of 2 × SDS loading buffer and boiled for 7 min. Input samples were mixed separately with 2 × SDS loading buffer and boiled as above. Samples were loaded on an SDS-PAGE gel and analyzed by immunoblotting. Immunoblotting was performed according to standard procedures.

For the peptide-Pex11 in vitro binding assay, 1 mg of each peptide was dissolved in 50 μL DMSO. Then, 1 μL of peptide solution was resuspended in 1 mL of ice-cold Binding buffer (50 mM Tris, pH 8.0 containing 50 mM NaCl, 5 mM EDTA) and incubated with EZview™ Red Streptavidin Affinity Gel (Millipore, Burlington, MA, USA, Cat#E5529) for 1 h at room temperature. After, the beads were washed (5× 1000× *g* for 1 min) to remove unbound peptides, 300 µL of the soluble fraction of protein lysate of certain His_6_-tagged GFP-Pex11s was loaded on each resin with bounded peptides, incubated for 1 h at room temperature, and then spun at 1000 g for 1 min to remove unbound proteins. The beads were washed five times with Binding buffer. Proteins were eluted as described above and analyzed by immunoblotting.

For the competitive in vitro binding, assay peptides were incubated with EZview Streptavidin agarose (Sigma-Aldrich, Cat# E5529) and washed as described above. Soluble fractions (300 µL) of His_6_-tagged GFP-Pex11 or His_6_-tagged GFP-Pex11 (45-249) were mixed with 150 pmol, 300 pmol, and 600 pmol of GST-PpPex19 or free GST, respectively, and loaded on each resin with bound H2 or H3 peptides. After 1 h of incubation at room temperature, samples were spun and washed five times with Binding buffer. Proteins were eluted as described above. Samples were loaded on an SDS-PAGE gel and analyzed by immunoblotting. 

### 2.5. Antibodies

Rabbit-anti-PpPEX19 antibody, rabbit-anti-PpPex3 antibody, rat-anti-PpPex2 antibody, and rabbit-anti-ScF1β antibody (which recognizes *P. pastoris* F1β) were custom made for the Subramani Lab. Living Colors^®^ A.v. mouse-anti-GFP (JL-8) antibody (Cat#632381) was purchased from Clontech (Mountain View, CA, USA). Mouse anti-GST Epitope Tag antibody (Cat#MMS-112R) was purchased from Covance (Princeton, NJ, USA). Pierce™ High Sensitivity Streptavidin-HRP (Cat#21130) was purchased from Thermo Fisher Scientific (Waltham, MA, USA). Peroxidase-conjugated, goat, anti–rabbit secondary antibody (Cat#172-1019) and peroxidase-conjugated, goat, anti–mouse secondary antibody (Cat#170-6516) were purchased from Bio-Rad Laboratories (Hercules, CA, USA). Peroxidase-conjugated, goat, anti-rat antibody (Cat#ab97057) was purchased from Abcam (Abcam, Cambridge, UK).

### 2.6. Peptide Binding Dot-Blot Assay

All peptides used in this work (Appendix A) were chemically synthesized by GenScript. Stocks (1 mg of each lyophilized peptide powder dissolved in 50 μL DMSO) were kept in −20 °C prior to use. In vitro dot-blot binding of Pex19 to the peptides was done as follows: peptides were diluted in DPBS, 2 μL of each solution containing 20 nmol or 5 nmol of peptide was then spotted on nitrocellulose membranes. After drying, peptide-containing cellulose membranes were blocked in 5% milk for 1 h at RT. Purified GST-Pex19 was then added to the membranes in a concentration of 20 μg/mL. As control, 20 μg/mL GST was added to the membrane. Binding of GST-Pex19 or free GST was detected immunologically by using monoclonal anti-GST or anti-PpPex19 antibodies. 

### 2.7. Protein Interactions in Y2H System

For Y2H analysis, the GAL4-based Matchmaker yeast 2-hybrid system (TaKaRa, Kusatsu, Shiga, Japan) was used. Full-length open reading frames and truncated or mutated forms were inserted in pGAD-GH (Clontech Laboratories, Mountain View, CA, USA, Cat#638853) or pGADT7-AD (Clontech Laboratories, Cat#630442) (AD), and pGBT9 (Clontech Laboratories, Cat#K1605-A) or pGBKT7 (Clontech Laboratories, Cat#630443) (BD) plasmids. The *S. cerevisiae* strain Y2H Gold was used (TaKaRa, Cat# 630498). For transformation of yeast, the LiAc-single-stranded carrier DNA-PEG method was used following the “Quick and Easy TRAFO Protocol” (http://diyhpl.us/~bryan/irc/protocol-online/protocol-cache/Quick.html, accessed on 1 February 2019). After transformation, yeast were spread on synthetic dropout (SD) medium (-LW) (Sunrise Science, Knoxville, TN, USA, Cat#1719) to select for transformants containing the introduced plasmids. Plates were incubated at 30 °C for up to 5 days. Next, three representative transformants from each strain were plated on proper selective media: SD medium (-LW) (Sunrise Science, Cat#1719), SD medium (-LWH) (Sunrise Science, Cat#1725) w/ 3-amino-1,2,4-triazole (Sigma-Aldrich, Cat#A8056) at the concentration indicated in the figures to test protein interactions. Results were analyzed after 3–7 days.

### 2.8. Fractionation and Carbonate Extraction

Cells were grown in YPD medium and then switched to methanol medium for 3 h. Three-hundred-and-seventy-five (375) OD of cells were pelleted and washed twice with water. Cells were resuspended in 3 mL of Zymolyase buffer (0.5 M KCl, 5 mM MOPS-KOH, pH 7.2, 10 mM Na_2_SO_3_, 12.5 mg Zymolyase-100 T/mL) and incubated for 30 min at 30 °C, 80 rpm. The cells were then spun down at 220 for 8 min at 4 °C and resuspended in 1.5 mL homogenization buffer (5 mM MES-KOH, pH 5.5, 1 M sorbitol, 5 mM NaF, 20 mM EDTA). Cells were lysed by applying 15 firm strokes in a Dounce homogenizer. The unbroken cell debris and nuclei were removed by two sequential centrifugations at 1000 g for 10 min at 4 °C. After the second centrifugation, the supernatants were considered to be the postnuclear supernatants (PNS). Centrifugation of the PNS at 20,000 g for 30 min at 4 °C (Optima Max-E, Beckman Coulter, Brea, CA, USA) generated supernatant (20S) and pellet (20P) fractions. The 20P samples were then aliquoted into four tubes. In one tube, the pellet was suspended in Tris buffer (10 mM Tris/HCl (pH 8)). In another, the pellet was placed in urea buffer (2 M urea, 10 mM Tris/HCl (pH 8)). The pellet in the third tube was in carbonate solution [100 mM sodium carbonate, 10 mM Tris/HCl, (pH 11.5)]. In the last tube, the pellet was dissolved in detergent solution (10 mM Tris/HCl (pH 8), 0.1% TritonX-100). Samples were incubated for 30 min at 4 °C. After that, samples were centrifuged at 200,000× *g* (Optima Max-E, Beckman Coulter) to generate high-speed supernatant (S) and pellet (P), which were analyzed by immunoblotting. 

### 2.9. Fluorescence Microscopy

*P. pastoris* cells were grown to late exponential phase in YPD medium, diluted to OD600 of 0.1 with fresh YPD, and grown to the early/mid-exponential phase. Next, cells were washed twice with YNB solution and inoculated into peroxisome proliferation (methanol) medium for 5 h or 16 h and immediately taken for microscopy observation. Images were captured at room temperature using a motorized fluorescence microscope (Axioskop 2 MOT, Carl Zeiss, Jena, Germany) with a Plan-Apochromat 100 ×/1.40 NA oil differential interference contrast (DIC) objective lens and monochrome digital camera (AxioCam MRm; all from Carl Zeiss). Optimal exposition times were automatically applied to capture images. All images were acquired and processed using the AxioVision software (Carl Zeiss), version 4.8.2.

### 2.10. In Silico Analysis

A sequence similarity search for the Pex11 protein was performed as a Standard Protein BLAST analysis on a dataset of the nonredundant (nr) protein sequences, using the default parameter and the algorithm blastp (protein-protein BLAST) at the National Center for Biotechnology Information (http://www.ncbi.nlm.nih.gov/BLAST/, accessed on 1 February 2019). The organism names, abbreviations, and GenBank accession numbers of Pex11 homologs are the following: Pp—*Pichia pastoris*, CAY69135; Hp—*Hansenula polymorpha*, DQ645582; Sc—*Saccharomyces cerevisiae,* CAA99168; Ca—*Candida albicans*, EAK92906; and Yl—*Yarrowia lipolytica*, CAG81724. A prediction of Pex19-BSs within Pex11 was done using the Pex19BS BLOCK prediction matrix containing both yeast and human Pex19-targeting elements [34,36] generated from Target signal predictor software (http://216.92.14.62/Target_signal.php, accessed on 1 February 2019). Multiple alignments of protein sequences were done using Multalin (http://multalin.toulouse.inra.fr/multalin/, accessed on 1 July 2020) [37] or Clustal X software package (http://www.clustal.org/clustal2/, accessed on 1 July 2020) [38]. Physiochemical properties like distributions of amino acids, hydrophobicity, hydrophobic moment, and amphipathicity assessments for α-helical peptides were predicted and projected in helical wheel diagrams by the HeliQuest (http://heliquest.ipmc.cnrs.fr/, accessed on 1 November 2021) [39].

### 2.11. Circular Dichroism Spectroscopy

Circular dichroism (CD) assays were performed on a Jasco J-815 spectropolarimeter at the Biophysics and Biochemistry Core of the Scripps Research Institute to assess secondary structure characteristics of each peptide. Peptides H3 and H4 were dissolved in 1× PBS pH 7.5 to reach 1 mg/mL concentration and peptides H2 and H2 L8A, Y10A in 1× PBS pH 7.5 with the addition of 0.5% DMSO to 1 mg/mL concentration. Peptides were supplied at 1 mg/mL and diluted to 0.1 mg/mL for analysis in 10 mM sodium phosphate, pH 7.4, 10 mM NaCl w/wo 0.5% DMSO, and w/wo addition of 30% TFE. A 0.1 cm quartz cuvette was filled with 200 μL of each sample, and the resulting signal was subtracted from the buffer control. Experimental scanning parameters were 270–185 nm, with 100 nm/min scanning speed, 1 s response, 5 nm bandwidth, and 1 nm data pitch. Compartment chamber was set to 25 °C. Each experiment was performed 10 times and data averaged, followed by analysis using the CAPITO web server (https://data.nmr.uni-jena.de/capito/index.php, accessed on 1 November 2021). Only data with an HT voltage below 550 is shown.

## 3. Results

### 3.1. P. pastoris Pex11 Contains a Classical Pex19-Binding Site Near Its N-Terminus

Like other PMPs, the transport and incorporation of Pex11 into the peroxisome membrane depend on Pex19 [40]. All Class 1 PMPs contain from one to several Pex19-BSs, whose position within Pex11 differs among species. In yeast, such as *Saccharomyces cerevisiae*, *Hansenula polymorpha,* or *Penicillium chrysogenum,* the Pex19-BS is located near the N-terminal end [36], while in mammalian Pex11 isoforms, it is near the C-terminus [24,41,42]. Thus, since some PMPs contain more than one Pex19-BS [23,31], we decided to precisely map it in PpPex11. 

The analysis of the 249 amino acid (aa) PpPex11 was initiated by inferring Pex19-BS motifs obtained from the target signal predictor website (http://216.92.14.62/Target_signal.php, accessed on 1 February 2019). Unfortunately, the prediction was ambiguous, as it did not indicate a single, strong candidate, but suggested instead, several putative Pex19-BS-like consensus sequences almost evenly distributed in the protein (see Appendix A for a graphic representation of predicted Pex19-BSs within PpPex11). 

Since Pex19-BS predictions are prone to yield a high level of false positives [43], the binding of Pex19 to regions of Pex11 was validated experimentally. We analyzed the Pex11–Pex19 interaction in vivo by generating a series of truncated forms of PpPex11 and testing their ability to bind Pex19 using the yeast two-hybrid (Y2H) assay. Several truncated forms of Pex11 did interact with Pex19, showing moderate to weak growth on selection media. These results pointed to the region between aa20-55 in PpPex11 as the segment recognized by Pex19 (Appendix A). This region contains the second helix (denoted H2) of the N-terminal, cytosolically-exposed domain of Pex11, which was proposed to bind Pex19, in yeast Pex11 proteins (Appendix A).

To further confirm and narrow the sequence in Pex11 recognized by Pex19, the truncation Pex11 (1-180) and the deletion mutant, Pex11 (Δ84-124) (both lacking different hydrophobic domains (HD) and previously showing interaction with Pex19), were subjected to site-directed mutagenesis within the H2 region. Both deletion of the 30-45 region (∆30-45), or substitution of a Leu by a Pro (at aa35) within these Pex11 forms, were sufficient to disrupt Pex11-Pex19 binding (Figure 2A), suggesting the presence of only one Pex19-BS within Pex11. 

Two short peptides corresponding to the H2 region and correlated with one of the predicted sites from the peptide scan (Appendix A), were used for in vitro dot-blot analysis with a GST-PpPex19 fusion protein. In this assay, membranes spotted with the peptides, including controls, were incubated with either the purified GST-Pex19 fusion protein or with GST alone. Among the synthetic peptides tested, only aa30-45 was recognized by GST-PpPex19, but not by GST alone, demonstrating the precise location of Pex19-BS in Pex11 (Figure 2B). 

As stated earlier, some PMPs contain multiple Pex19-BSs within their sequence. Because such regions are often difficult to test in the Y2H system [44], we confirmed whether binding of Pex19 to Pex11 happens also using an in vitro pull-down assay. 

In Pex11 (45-249), the deletion of aa1-44 was sufficient to completely abolish Pex11-Pex19 complex formation (Figure 2C). The full-length Pex11 (Pex11) and its C-terminally-truncated form, (Pex11 (1-180)), bound Pex19 in the assay, showing that Pex11 directly interacts with Pex19 only via the Pex19-BS located in its H2 region.

### 3.2. Mutational Analysis of the Pex19-BS of PpPex11 and Its Recognition by Pex19 

Pex19 recognizes Pex19-BSs in PMPs via its globular C-terminal domain (Pex19 CTD) [45,46], which forms a helical bundle with high sequence conservation within the first helix of the CTD, named helix alpha-1 [47,48]. The helix generates an extensive hydrophobic patch essential for Pex19-BS recognition in PMPs. Because most of the Pex19-BSs described so far consist of a cluster of basic and hydrophobic residues (usually with Leu in the center [49]) [31,36,50,51], we asked which residues of H2 in Pex11 are critical for Pex19 binding. 

Pex11 H2 variants were generated with mutations within the hydrophobic or hydrophilic sites of the helix, but without disturbing the amphipathicity of H2 (Figure 3B,C), and subjected to dot-blot analysis (Figure 3). For instance, in the hydrophobic site, we mutated those amino acids that differentiate Pex11 from HsPex11β because the H2 helix of HsPex11β does not bind Pex19. 

No Pex19 binding was detected to H2 of HsPEX11β, as expected, or to H2 L8A, or to other H2 peptides with the mutation in position 8 (i.e., H2 muts and H2 L8A, Y10A peptides) of PpPex11 (Figure 3A). This result suggests that L8 is critical for the binding of H2 in PpPex11 to Pex19. Furthermore, substitution of hydrophobic Tyr (aa10) or Leu (aa7), as well as the positively-charged residue, Lys (aa1), also reduced Pex19 binding ability, demonstrating that that these residues also contribute to the recognition of PpPex11 by Pex19. Interestingly, substitution of Tyr (aa18) or both Tyr and Leu (aa4) to positively-charged Lys improved Pex19 binding ability. The GST control did not bind at all to any of the peptides on the membrane as anticipated (not shown).

We also analyzed whether mutations within the H2 peptide that disrupt Pex19 binding (e.g., the H2 L8A, Y10A peptide) affect the predisposition of this peptide to fold into an α-helix. Both wild-type H2 and H2 L8A, Y10A peptides were analyzed by circular dichroism (CD) spectroscopy. As it has been observed previously for other peptides for which amphipathic helix structures were predicted [52], the addition of the secondary structure inducer, 2,2,2-trifluoroethanol (TFE), induced these peptides to adopt α-helical structures, as reflected by significant changes in the CD spectra. However, neither peptide (H2 and H2 LA8, Y10A) formed any secondary structures in aqueous solution (Figure 3C). These data show that the Pex11 H2, as well as the mutant H2 L8A, Y10A peptides defective in Pex19 binding do indeed form α helices. 

### 3.3. Pex19 Chaperones Pex11 by Binding to Its Pex19-Binding Site

Pex19 acts as a chaperone for many PMPs in addition to its function as their import receptor [23,28,30,32,46]. Supporting this chaperone role are observations that many PMPs are unstable in the yeast *pex19*Δ strain [28]. Additionally, in human cells lacking peroxisomes resulting from Pex3 deficiency, the expression of Pex19 can significantly extend the half-lives of PMPs [23]. Since many PMPs contain multiple Pex19-BSs that are located within or close to their transmembrane domains (TMDs), it has been suggested in a current model, but not proven experimentally, that Pex19 binding to these regions allows it to function as a chaperone by masking TMDs of newly-synthesized PMPs, thereby preventing their aggregation and clearance [23,28,32,46]. However, contrary to the PMPs tested in these studies, Pex11 has only one Pex19-BS, located far from its TMDs. 

Therefore, we tested whether Pex19 chaperone activity also stabilizes Pex11. Due to the lack of good Pex11-specific antibodies, we used various Pex11 fusion constructs. First, we compared levels of the GFP-Pex11 fusion protein expressed from its endogenous or constitutive promoters in WT and *pex19*Δ strains at different time points after methanol induction of peroxisomes. We included a constitutive promoter because the level of peroxisomal proteins can be reduced in the absence of peroxisomes. Immunoblot analyses revealed that independent of the promoter that was used to control GFP-Pex11 expression, it was much more abundant in WT cells that express Pex19, than in cells lacking Pex19 (Figure 4A). The levels of GFP-Pex11 in *pex19*Δ, but not in WT, strains gradually decreased over time, even when a constitutive promoter was used. Thus, differences in GFP-Pex11 levels between WT and *pex19*Δ cells cannot be simply explained by alterations in transcription initiation, but rather reflect the instability of GFP-Pex11 and its degradation. 

In the absence of Pex19, some mislocalized PMPs expose hydrophobic surfaces in the cytoplasm, which are then detected and removed by the AAA-ATPase Msp1 [53], target mislocalized, tail-anchored (TA) proteins [54]. We asked whether the observed decrease in Pex11 levels is solely a consequence of decreased Pex11 half-time in the absence of Pex19, or if it is caused by other factors that eliminate mistargeted Pex11. We compared Pex11-2xHA endogenous levels in *pex19*Δ and *pex3*Δ cells, which have no functional peroxisomes, as Pex3 anchors Pex19 at the peroxisomal membrane and assists in PMP insertion [25,47,55,56,57]. For this comparison, we eliminated the pexophagic degradation of peroxisome remnants and components, using *atg30*Δ cells as the background. While the Pex11-2xHA levels decreased over time in *pex19*Δ cells, they remained stable in the *pex3*Δ strain (Figure 4B), ruling out the activation of quality control mechanisms by Pex11 mistargeting.

Finally, to determine whether indeed Pex19 stabilizes Pex11 through physical interactions mediated by the Pex19-BS in Pex11, we compared levels of GFP fusions of full-length and truncated forms of Pex11 lacking the Pex19-BS, in cells expressing Pex19. As a control, we also tested the Pex11 (1-223) truncated form and free GFP. The absence of the Pex19-BS in the GFP-Pex11 fusion (GFP-Pex11 (45-249)) rendered it highly unstable in WT cells. After peroxisome induction, the levels of full-length GFP-Pex11 and GFP-Pex11 (1-223) increased over time, while that of GFP-Pex11 (45-249) rapidly decreased (Figure 4C). We excluded pexophagy involvement in Pex11 (45-249) degradation using a pexophagy-deficient *atg30*Δ strain (Appendix A). Thus, consistent with the previously-documented stabilizing effect of Pex19 on other PMPs, direct binding of Pex11, via its Pex19-BS, to Pex19 prevents Pex11 from premature degradation. This must then raise the question regarding how Pex19 chaperones Pex11.

### 3.4. Binding Pex19 to Pex11 Prevents Pex11 Oligomerization

Pex11 engages in homotypic protein–protein interactions, which in addition to being a requirement for Pex11 function in catalyzing peroxisome division, could also cause its non-specific aggregation in the cytosol. Several studies proposed that self-interaction of Pex11 is caused by the H3 region (and H2 in human) [58,59]. To further shed light on Pex19 chaperone function, we asked whether the Pex19 binding to Pex11 in close proximity to H3 might prevent Pex11 oligomerization and further cytoplasmic aggregation of Pex11.

We first determined which helices of Pex11 are required for its oligomerization. We noticed that removal of regions of H2 (Pex11 (1-98, Δ30-45)), H3 (Pex11 (1-98, Δ55-86)), or substitution of a Leu^35^ by a Pro (within the H2 segment, Pex11 (1-98, L^35^P)) completely abolished Pex11 dimerization (Figure 5A), showing that both H2 and H3 helices contribute to Pex11 dimerization. 

We further validated and quantified the roles of H2 and H3 in dimerization by in vitro pull-down assays. Biotinylated H2 and H3 peptides were immobilized on streptavidin-coated agarose beads and used to pull down His_6_-GFP-Pex11 (Figure 5B). Interestingly, Pex11 was pulled down better by the H2, than by the H3, peptide (Figure 5B). We did not observe any interaction between Pex11 and the H4 peptide (used as a negative control), which corresponds to another amphipathic helix within Pex11 that we identified during our in silico analysis done by Heliquest software, or to streptavidin-coated beads without any peptide (negative control or C^−^ in Figure 5B). This result was unexpected, because H3 was predominantly assumed to enable Pex11 oligomerization [60]. However, our pull-down assay showed that His_6_-GFP-Pex11 bound to the H2 peptide was ∼2.5-fold better than H3 in vitro. We next tested the effect of N-terminal truncation on the Pex11 self-interaction to further clarify the role of H2 in dimerization. As shown in the quantification of the Western blot signals in Figure 5C, deletion of aa1-44, which includes H2, dramatically reduced the binding of Pex11 (45-249) to both H2 and H3. Interestingly, homotypic H3-H3 interactions were the weakest among all tested combinations, suggesting that H2 plays a critical role in the homodimerization of Pex11, whether by formation of H2-H2 or via H2-H3 and/or H3-H2 interactions (Figure 5D). We also noticed that H2 and H3 contribute additively to Pex11 dimerization, because the quantified binding affinity of Pex11 can be presented as a sum of the contributions of each helix to the interactions between them. 

Since the H2 region of Pex11 has both the Pex19-BS and dimerization property, we subjected mutants of this peptide described in Figure 3 to pull-down with full-length Pex11. Single mutations of either residues from the hydrophobic face of H2 to a much smaller residue Ala (L7A, L8A, L11A and Y18A), or positively-charged residues to Asn (R1N, K3N, R6N, and K13N) in the hydrophilic site, did not abolish the interaction between the H2 peptide and Pex11. Reduced binding was visible for the H2 Y18A and Y10A mutants with the Ala substitution in the core of the hydrophobic interface, and for the H2 single or double-mutant L4K, Y18K. Only the peptide H2 muts with four Ala substitutions on its hydrophobic face, and the peptide H2 L8A, Y10A with two Ala substitutions on its hydrophobic face, completely failed to bind Pex11 (Figure 6A). The corresponding H2 peptide of HsPex11β also showed significantly reduced interaction with Pex11 protein in vitro. These results show that individual Leu or Tyr residues make only minor contributions to the Pex11 self-interaction, but the whole hydrophobic face of H2 with many van der Waals contacts plays a role in this interaction. Since H2 mutants R1N and K3N showed reduced affinity toward Pex11, we conclude that dimerization can be additionally supported by H-bond or salt–bridge interactions of charged residues flanking the hydrophobic interface. 

Since H2 of Pex11 harbors its Pex19-BS and the residues for Pex11 H2-dependent dimerization, we hypothesized that Pex19 and Pex11 might competitively occupy the same surface of the H2 helix. We asked if Pex19 competes with Pex11 for access to the H2 peptide, and also whether Pex19 binding to Pex11 affects Pex11 capture by the H3 peptide. Our pull-down competition assays (Figure 6B) showed that GST-Pex19 binds to H2, but not to the H3 peptide, in the presence of Pex11. Additionally, increasing the amount of GST-Pex19, but not GST alone, prevented Pex11 from binding to the H2, as well as to the H3 peptides, showing that Pex11 binding to Pex19 and Pex11 oligomerization are mutually competitive. We did not observe any effect of increasing Pex19 levels on pulling down Pex11 (45-249), which is defective in Pex19 complex formation due to the lack of the Pex19-BS, which suggests that Pex11–Pex19 complex formation does indeed prevent Pex11 self-interaction, thereby providing insight into how Pex19 functions as a chaperone.

### 3.5. Pex11 Insertion into the Peroxisomal Membrane, but Not Its Transport to Peroxisomes, Depends on Physical Interaction between Pex11 and Pex19

Snyder et al. [30] proposed that, at least in *P. pastoris*, Pex19 does not function only as a PMP import receptor, but rather acts as a chaperone that facilitates the insertion of PMPs into peroxisomal membrane. Thus, we asked if Pex19 is necessary for the targeting of Pex11 to peroxisomes, by comparing the subcellular localization of Pex11 (45-249) (lacking the Pex19-BS), with that of full-length Pex11. 

GFP-tagged Pex11 (45-249) or full-length Pex11 constructs were expressed from their endogenous *PEX11* promoter in the methanol-grown *pex11*Δ strain, which also produces mPTS-RFP to mark peroxisomes. The *pex11*Δ strain was used to avoid potential dimerization of the introduced fusion protein with endogenous Pex11, which could affect protein localization. Fluorescence microscopy analysis demonstrated that deletion of the Pex19-BS in Pex11 did not lead to a complete loss of peroxisomal targeting. Although some impairment in Pex11 (45-249) targeting peroxisomes could be noticed, still significant co-localization of GFP with mPTS-RFP was visible (Figure 7A). This result indicates that Pex11 can be transported to peroxisomes even when its interaction with Pex19 is abolished. Since we previously excluded the existence of other Pex19-BSs within Pex11 (Figure 2C), this result shows that the Pex19-BS, which by itself contains a mPTS, and the novel, Pex19-independent mPTS of Pex11 are separable, and that Pex19-independent peroxisomal targeting driven by the novel mPTS (described in next section) is distinct from the role of the Pex19-BS in membrane insertion of Pex11, which is clarified next. 

The fluorescence studies left unclear whether Pex11 (45-249) is only targeted to peroxisomes or is also inserted into the membrane. Protein membrane extraction was performed on whole peroxisomes isolated from methanol-grown, *pex11*Δ cells expressing the GFP-Pex11 (45-249) fusion protein. In parallel, we confirmed that full-length Pex11 is an integral membrane protein, as expected. The effects of alkaline (pH 11.5) sodium carbonate and urea extraction on separating peripheral from integral membrane proteins are widely described. However, because some peroxisomal peripheral membrane proteins are still somehow resistant to sodium carbonate or urea treatment, we established a concentration of Triton X-100, at neutral pH, that releases peripheral membrane proteins into the supernatant, without affecting integral membrane proteins, which are still retained in the insoluble membrane pellet. In all analyzed samples, the integral membrane marker, Pex3, was in the pellet after urea, sodium carbonate, or Triton X-100 treatment (Figure 7B, lower lane). Functional Pex11-2xHA, as well as GFP-Pex11, behaved similarly to integral membrane proteins, even in the presence of Triton X-100. However, the GFP-Pex11 (45-249) fusion protein was completely released into the supernatant fraction after treatment with Triton X-100 at low concentration, demonstrating that Pex11 (45-249) was not incorporated into the peroxisomal membrane like full-length Pex11 or Pex3 (Figure 7B). However, Pex11 (45-249), even as a peripheral membrane protein, was still tightly associated with the peroxisomal membrane based on its insolubility in sodium carbonate and urea (Figure 7C). 

### 3.6. Mapping of the mPTSs of Pex11

In order to delineate the mPTS of Pex11, various GFP-tagged, truncated Pex11 proteins were generated and transformed into the *pex11*Δ strain, expressing mPTS-RFP as a peroxisomal marker. The sub-cellular localizations of the fusion proteins were determined by fluorescence microscopy 5 h after the induction of peroxisome proliferation in methanol medium (Figure 8). All analyzed truncated forms of Pex11 tested co-localized with peroxisomes. Even the shortest fragment, containing aa156-249 of Pex11 (Pex11 (156-249)), was sufficient to colocalize with peroxisomes, although its expression was low and some other cytosolic spots were visible. 

This region of Pex11 contains a previously-defined phosphorylation site required for its interaction with the fission machinery protein, Fis1 [61]. To examine the role of this modification on the Pex19-independent mPTS, we generated the Pex11 (45-249, S^173^A) and Pex11 (45-249, S^173^D) mutants, mimicking the constitutively-unphosphorylated (S173A) and constitutively-phosphorylated (S173D) status of Pex11 (45-249), respectively. Fluorescence microscopy analysis demonstrated that both phospho-mutants of GFP-Pex11 (45-249) colocalized with mPTS-RFP (Appendix A). The peroxisomal localization of GFP-Pex11 (45-249, S^173^A) and GFP-Pex11 (45-249, S^173^D) fusion proteins indicates that the Pex19-independent mPTS activity of Pex11 is not dependent on its phosphorylation at S173 and Fis1 binding.

Other deletion mutants were generated, where aa162-200 or aa203-218 were removed from Pex11 (Pex11(Δ162-200) and Pex11 (Δ203-218), respectively), or from Pex11 (45-249) variants (Pex11 (45-249, Δ162-200) and Pex11 (45-249, Δ203-218), respectively), and tested for their co-localization with the mPTS-RFP in the *pex11*Δ strain. Only the lack of aa203-218 interfered with the peroxisomal targeting of GFP-Pex11 (45-249) (Figure 9A and Appendix A), demonstrating that this region has another mPTS. Additionally, the targeting of Pex11 (156-249), containing this region, to peroxisomes, showed that it was also sufficient for peroxisomal localization, even though this protein was unstable (Figure 8). 

The analogous truncated fusion protein lacking this second mPTS, but containing the Pex19-BS (Pex11 (Δ203-218)), co-localized strongly with peroxisomes, pointing to the presence of another mPTS in Pex11, likely within the Pex19-BS (Figure 9A). 

We confirmed that aa203-218 of Pex11 containing the novel mPTS does not bind Pex19 directly (Figure 9B, peptides 3–7). Finally, our in silico analysis revealed that aa203-218 is well preserved among Pex11 proteins and forms a putative amphipathic α-helix structure, which we named H4 (Figure 9B). To analyze whether H4 can fold into an α-helix, we analyzed this peptide by CD spectroscopy, as we did for the H2 peptide (Figure 3C). The H3 peptide was used here as a reference control, since it is known to have amphipathic helix properties [52]. Indeed, the addition of TFE caused the folding of both peptides (H3 and H4) into an α-helix (Figure 9C). These data demonstrate that the H4 region also has the potential to fold into an α-helix.

All together, these results show that Pex11 contains a Pex19-independent mPTS in H4 that can target Pex11 to peroxisomal membranes. It is noteworthy that this new mPTS corresponds to a recently described amphipathic helix of HsPex11γ, which is suggested to be anchored in the outer leaflet of the peroxisome membrane [22]. Because this region was suggested to also mediate interactions with other Pex11 isoforms in humans, we tested its involvement in Pex11 self-interaction. However, neither the Pex11 (H4) peptide alone, nor its removal from Pex11 (Pex11 (Δ203-218)), impaired Pex11 binding (Appendix A).

## 4. Discussion

The Pex11-family of proteins is required for peroxisome division and the maintenance of peroxisome number. Uncovering how these proteins are targeted and incorporated into the peroxisomal membrane lies at the basis of our understanding how peroxisomes grow and divide. We present a detailed analysis of the *P. pastoris* Pex11 protein and its interaction with the chaperone Pex19, which is also required for Pex11 insertion into peroxisomal membrane, but not essential for its trafficking to the peroxisome surface. Here we demonstrate that, in Pex11, two mPTS sequences exist, and each of these motifs is sufficient for Pex11 trafficking to the peroxisomal membrane, but while one depends on Pex19 binding, the other does not.

### 4.1. Recognition of the Pex19-BS of Pex11 by Pex19

PMP binding maps to the PEX19 CTD to the region comprising the α1 helix and a lid region [48]. These two regions together form a groove on the PEX19 CTD surface with two cavities that are large enough to accommodate the Pex11 hydrophobic face comprised of the aromatic and/or aliphatic side chains of its Pex19-BS, if an α-helical conformation is adopted [41]. Indeed, the Pex19-BS of Pex11, especially its H2 region, adopts an α-helical conformation, as shown here, and exhibits a hydrophobic patch comprised of aromatic or aliphatic side chains that could fit into hydrophobic pockets formed on the Pex19 CTD surface.

We validated, experimentally, the residue(s) within the Pex19-BS region that are critical for Pex11 binding to Pex19. Our data revealed that while the hydrophobic patch on the Pex19-BS helix of Pex11 is critical for the interaction (especially L8 and Y10 from the middle of the hydrophobic segment), additional binding comes from positively-charged residues residing adjacent to the hydrophobic surface. Moreover, the binding efficiency of the Pex11(H2) peptides to Pex19 is further improved if the flanking, hydrophobic residues (aa4 and aa18) are substituted by Lys.

### 4.2. Chaperone Activity of Pex19 Is Mediated by Competitively Preventing Pex11 Oligomerization

We demonstrate that Pex19 protects Pex11 from degradation, and the chaperone role of Pex19 explicitly requires its direct interaction with the Pex19-BS in Pex11 to maintain the stability of Pex11. It has been proposed that the chaperone activity of Pex19 depends on its ability to bind hydrophobic domains in PMPs that typically lie within their Pex19-BSs, as such binding could disrupt interactions between hydrophobic domains and prevent PMP aggregation in the cytosol [23]. However, the Pex19-BS of Pex11 is located far from its HD1 and HD2 domains, but close to the amphipathic helix H3 believed previously to be involved in self-interaction and peroxisomal membrane elongation [58]. We show here for the first time that Pex11 can self-interact in vitro using homo- and/or heterotypic interactions involving both its H2 and H3 helices, and the binding mediated by H2 is much stronger than that mediated by H3. Our data revealed that Pex19, by occupying Pex19-BS located within H2, prevents dimerization of Pex11, as the same interface is required for both interactions. This explains how Pex19 functions as a chaperone for Pex11.

Although evidence for Pex11 oligomerization is found in several reports, how oligomerization of Pex11 is achieved has not been determined so far. Our results, therefore, open new directions for mechanistic insights of Pex11-dependent peroxisome division and Pex19’s contribution to peroxisome maintenance. It is particularly interesting how these two processes (Pex11 oligomerization and Pex11-Pex19 binding) are implicated in membrane remodeling activity of Pex11, since a pool of Pex19 associates with the peroxisomal membrane.

### 4.3. Pex19-Dependent and -Independent mPTSs in Pex11

Previous results show clearly that in both yeast and human PMPs, there exist examples where the mPTS and Pex19-BSs do, or do not, overlap [30,33]. Our results are consistent with the idea that Pex11 has two mPTSs, one that is Pex19-dependent and overlaps with its Pex19-BS, and the other being Pex19-independent and incapable of interacting with Pex19.

### 4.4. Pex19 Has a Distinct Role in Chaperoning and Membrane Insertion of Pex11, and in Peroxisome Targeting

Because the majority of PMPs have mPTSs that overlap with, and have not been dissected from, Pex19-BSs in their vicinity, it has been suggested that Pex19 is critical for peroxisomal targeting and membrane insertion of PMPs [36]. However, our results are more in agreement with our previous report suggesting that Pex19 may not function as a sole targeting receptor, for at least some PMPs [30].

Precise mapping of the Pex19-dependent and the Pex19-independent mPTSs in Pex11 allowed us to prove that Pex19 does not bind the latter region located in the H4 region, which is sufficient for peroxisomal targeting of the Pex11 (45-249) mutant (lacking the Pex19-BS). These results suggest that Pex11, unlike the majority of PMPs, can use two mechanisms of transport to peroxisomes, where only one of them depends on its direct interaction with Pex19 protein, and the other does not.

The Pex19-independent targeting of PMPs to the peroxisome membrane raises the question as to the mechanism of such targeting. Pex3 targeting to peroxisomes is already known to be Pex19-independent. It needs further investigation to uncover if Pex19-independent targeting of Pex11 to the peroxisomal membrane depends on other proteins that act as the mPTS receptor at the peroxisome membrane. In this context, it is widely accepted that tail-anchored proteins can be targeted to particular cellular compartments based on the length of their TMD and their flanking amino-acid composition [62]. Unfortunately, the mPTSs of only a few PMPs have been defined so far, including that of Pex3, which is Pex19-independent in its targeting to peroxisomes, and no amino acid consensus has been defined for an mPTS [63,64,65].

We also considered the possibility that Pex11 uses a piggyback mechanism to reach peroxisomes, as demonstrated for the transport of peroxisomal matrix proteins [66], but this has not been documented for PMPs. By analogy, the Pex19-independent mPTS we identified in Pex11 might serve as a binding site for another protein targeted to the peroxisomal membrane by the Pex19 receptor. The Pex11 (45-249) form that lacks a Pex19-BS still associates with peroxisome membranes (Figure 7). Similar observations have been reported for the mPTSs of human PEX13, *Candida boidinii* PMP47 and *P. pastoris* Pex3, and it has been suggested that these mPTSs are tightly anchored to the peroxisomal membrane via another unknown PMP [33,65,67]. Pex11 interacts with several other PMPs, including but not limited to: Fis1 [22], Pex14 [68,69,70,71], and Pex34 (Pex36 in *P. pastoris*) [72]. We already excluded Fis1 involvement in Pex11 trafficking (Appendix A) but do not know if the other PMPs that interact with Pex11 could serve as a receptor for its targeting to peroxisomes.

### 4.5. Relevance of Our Findings to PEX11 Mutations Found in Human Patients

Our results are relevant in the context of patients with various mutations in the *PEX11β* gene (PEX11B c.64C>T p.(Gln22Ter) homozygous variant, PEX11B c.235C>T p.(Arg79Ter) homozygous; PEX11B c.136C>T p.(Arg46Ter) homozygous; PEX11B c.595C>T p.(Arg199Ter) heterozygous, PEX11B ex1-3 del heterozygous) [73,74]. All these patients manifest defects in peroxisome division, but without significant alteration of peroxisomal biochemical parameters, suggesting that not just metabolic aberrations contribute to the pathology of this peroxisome biogenesis disorder (PBD). These studies emphasize that peroxisome morphology is also an important factor contributing to human health. The described cases are assigned to the milder end of the disease spectrum, with congenital cataract as a consistent primary presenting feature, often accompanied by the later manifestation of other visual problems, mild intellectual disability, progressive hearing loss, sensory nerve involvement, gastrointestinal problems, and recurrent migraine-like episodes [73,74,75]. Interestingly, all identified biallelic loss-of-function mutations in *PEX11β* cause complete lack of PEX11β, in such a way that even truncated forms of PEX11β are not detected in these patients. It did not escape our attention that these mutations precede, or are within, the PEX19-BS in PEX11β (aa186-211) [41]. PEX19 is a predominantly cytoplasmic, partly peroxisomal protein [24,26,76] that stabilizes newly-synthesized PMPs in the cytoplasm by directly interacting with them [23]. Our data show that Pex19 has Pex11 chaperone activity when it binds to the Pex19-BS in Pex11 (Figure 4 and Appendix A). Since human PEX11s, contrary to their yeast homologs, contain the PEX19-BS closer to their C-termini, the patient mutations within PEX11β causing premature termination at the sites preceding the PEX19-BS would render these proteins unstable in the absence of Pex19 chaperoning activity.

## Figures and Tables

**Figure 1 cells-11-00157-f001:**
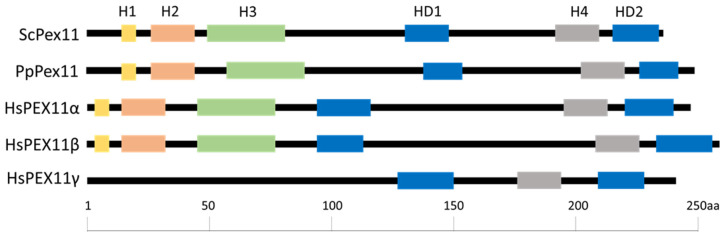
Graphical representation of the Pex11 protein in yeast and humans. Graphical representation of the domain/motif architecture organization for each Pex11 protein in yeast *S. cerevisiae*, *P. pastoris* and in humans. Proteins are drawn to scale with the amino acids (aa) number indicated at the bottom. Most, but not all (e.g., HsPex11γ), Pex11-family proteins have four putative amphipathic helices, whose coordinates in PpPex11 are as follows—H1 (aa14-19), H2 (aa25-45), H3 (aa55-86), and H4 (aa203-218), shown as yellow, orange, green, and grey rectangles, respectively) and two hydrophobic domains (HD1 and HD2, blue rectangles).

**Figure 2 cells-11-00157-f002:**
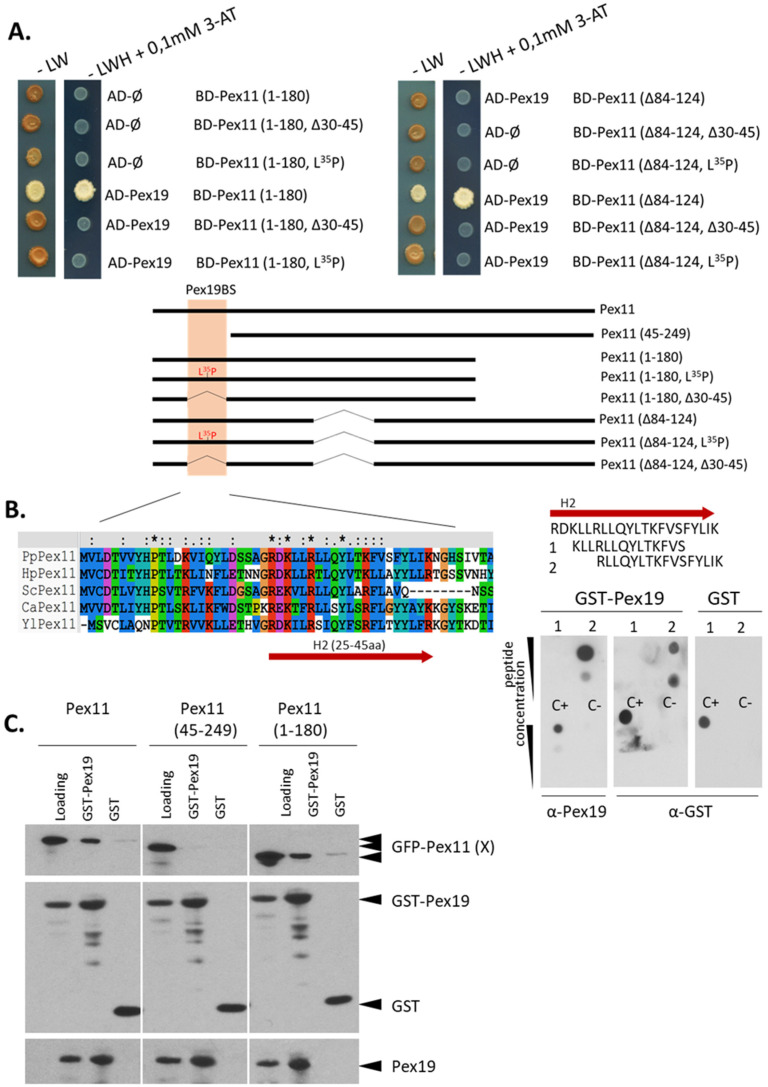
PpPex11 contains one Pex19-binding site (Pex19-BS) near its N-terminal end. Pex11 interacts with Pex19 via its Pex19-BS. (**A**) Y2H screen to determine the location of the Pex19-BS within PpPex11. Two truncated forms of PpPex11 were subjected to site-directed mutagenesis within H2 region of Pex11 and fused with the BD domain of GAL4 to test their abilities to interact with AD-Pex19. Schematic representation of Pex11 mutants used in this study is shown and the H2 helix (aa25-45), within which the Pex19-BS was mapped for other yeast Pex11s, is highlighted in orange. 3-AT, 3-amino-1,2,4-triazole; AD, activation domain; BD, DNA binding domain; -LW, yeast synthetic drop-out medium without leucine and tryptophan serving as a positive control to show equal plating of cells; -LWH, yeast synthetic drop-out medium without leucine, tryptophan, and histidine. (**B**) Precise mapping of the Pex19-BS. Based on the location of the Pex19-BS in the H2 helix of Pex11, 14-mer (aa27-40) and 16-mer (aa30-45) peptides, with a 3aa shift between them, were synthesized, spotted onto a nitrocellulose membrane, and subjected to the in vitro Pex19 binding assay. Peptides and proper positive (C^+^) and negative (C^−^) controls were spotted at two concentrations (20 nmol and 5 nmol, respectively) and tested for interaction with GST-PpPex19. Bound protein was detected immunologically with polyclonal anti-Pex19 or anti-GST antibodies. As a control, free GST and monoclonal anti-GST antibodies were used. Sequence alignment of N-terminal regions of Pex11 proteins from various species, showing conservation of specific residues within the H2 helix, is included. Residues in H2 helix are colored by Clustal X [38] based on their physico-chemical properties: hydrophilic, charged: D, E (magenta), K, R, H (red); hydrophilic, neutral: S, T, Q, N (green); hydrophobic: A, V, L, I, M, W, F (blue); P (yellow); G (orange; and other aromatic Y and H (cyan). Abbreviations and accessions numbers used in sequence alignments: Pp—*Pichia pastoris*, CAY69135; Hp—*Hansenula polymorpha*, DQ645582; Sc—*Saccharomyces cerevisiae*, CAA99168; Ca—*Candida albicans*, EAK92906; Yl—*Yarrowia lipolytica*, CAG81724. (**C**) The Pex11 (45-249) deletion mutant does not bind Pex19 in vitro. Binding studies revealed that removal of N-terminal end of Pex11 (aa1-44) is sufficient to inhibit Pex11–Pex19 binding in vitro. Only the mutant, Pex11 (45-249), was deficient in Pex19 binding, whereas the full-length and other deletion mutants, Pex11 (1-180), were still pulled down with GST-Pex19. Free GST protein was used as a control. There was equivalent loading in the input and bound lanes. Proteins were detected by anti-GFP, anti-GST, and anti-Pex19 antibodies.

**Figure 3 cells-11-00157-f003:**
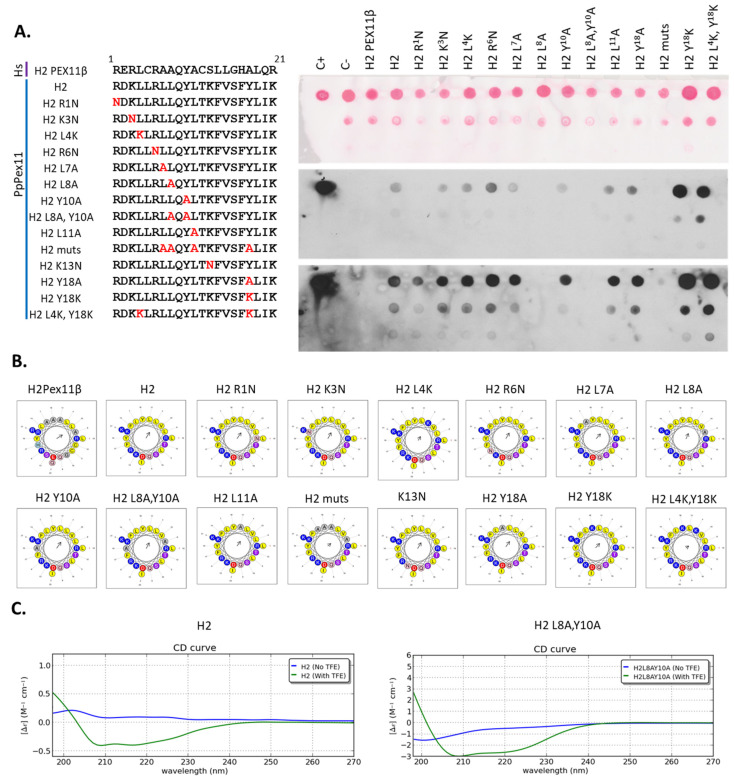
The Pex19-BS of Pex11 binds to Pex19 via its hydrophobic surface and flanking positively-charged residues. Identification of H2 mutants of Pex11 deficient in Pex19 binding by dot-blot analysis. (**A**) The H2 peptides generated, including one harboring the HsPex11β H2 sequence but lacking Pex19-binding properties, as well as the PpPex11 H2 WT or mutant sequences (sequence details are indicated and mutated residues are marked in red) are shown on left. Dot-blot analysis (shown on right) was done as described in Figure 2B. Peptides were spotted on a nitrocellulose membrane at two concentrations (20 nmol and 5 nmol) and tested for interaction with GST-PpPex19. The top panel shows spotted peptides stained with Ponceau S prior to GST-PpPex19 addition. Bound protein was detected immunologically with polyclonal anti-Pex19 antibodies in the middle (short exposure) and bottom (long exposure) panels. (**B**) Helical-wheel diagrams of HsPex11β and PpPex11 H2 variants using HeliQuest. Positively-charged residues are shown in blue, negatively-charged residues in red, and hydrophobic residues in yellow. In addition, Ser and Thr are shown in purple, Gly and Ala in gray, Asn and Gln in pink, and His in sky blue. The arrows represent the helical hydrophobic moment. (**C**) CD spectra of the PpPex11 H2 wild-type and mutant peptides. The spectrum shows that H2 and its mutant variant H2 L8A, Y10A are unstructured in phosphate buffer, but the addition of 30% of TFE induces changes in the spectrum, typical for α-helical structures.

**Figure 4 cells-11-00157-f004:**
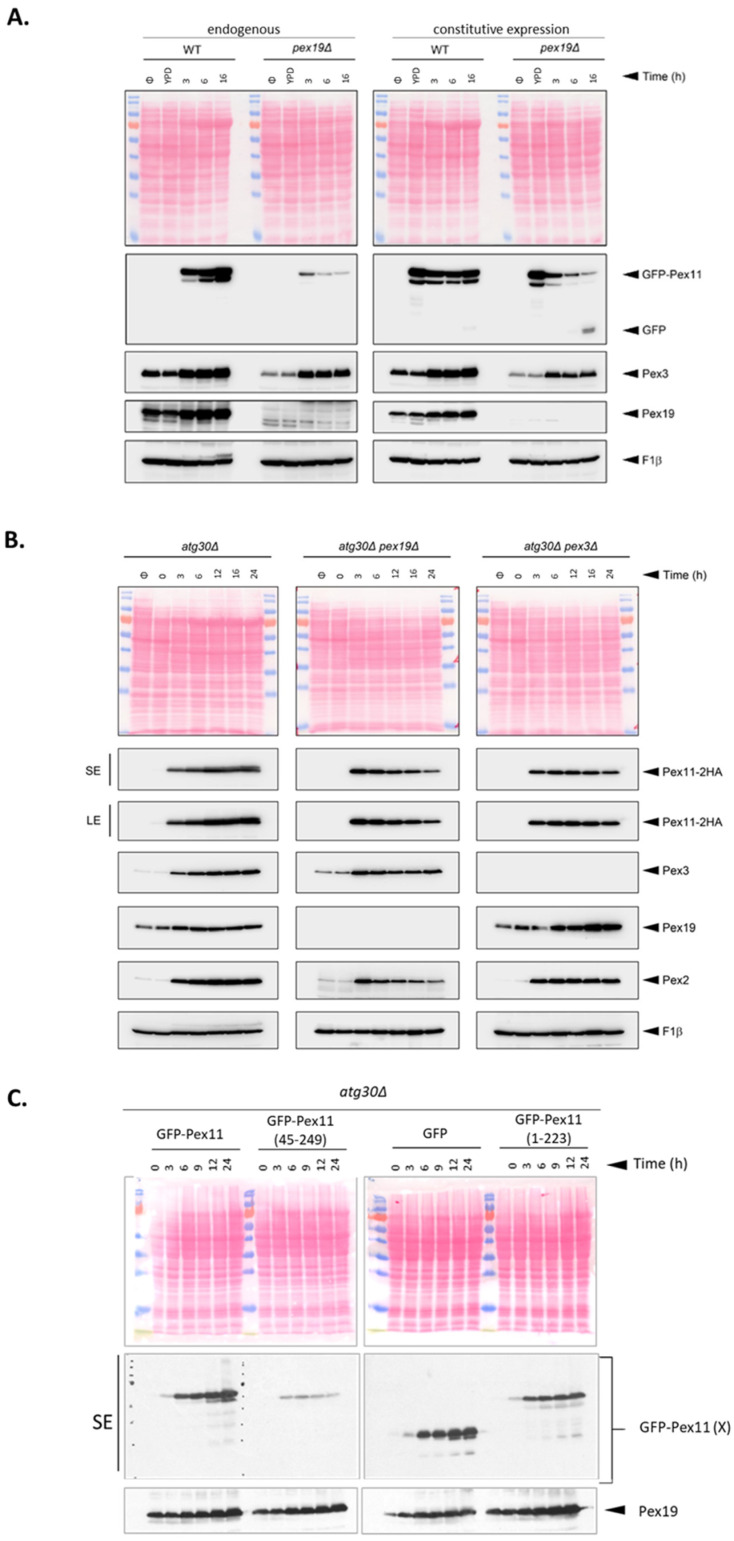
Pex11 is chaperoned by Pex19, which determines its stability via direct interaction with the Pex19-BS of Pex11. (**A**). Western blot analyses of GFP-Pex11 and other protein levels in WT and *pex19*Δ strains after methanol induction. WT and *pex19*Δ cells expressing Pex11-2HA from its endogenous *PEX11* promoter, or the constitutive *GAPDH* promoter, were grown in methanol medium, and 2 OD cells were collected at the indicated time points. GFP-Pex11 levels were visualized by anti-GFP antibodies. Pex3 and Pex19 were detected using custom antibodies, and F1β was used as loading control. (**B**). Western blot analysis of Pex11-2HA and other protein levels at various time points in peroxisome-deficient strains after methanol induction. WT, *pex19*Δ, and *pex3*Δ cells in an *atg30*Δ (pexophagy-deficient) background expressing Pex11-2HA from its endogenous promoter were grown in methanol medium, and 2 OD cells were collected at indicated time points. Endogenous PMPs were detected with indicated antibodies, and F1β was used as loading control. SE—short exposure and LE—long exposure. (**C**). Removal of Pex19-BS destabilizes Pex11. Western blot analysis of levels of truncated forms of GFP-Pex11 at various time points in atg30Δ cells after methanol induction. GFP-Pex11, GFP-Pex11 (45-249), GFP-Pex11 (1-223) (referred to collectively as GFP-Pex11 (X)), and free GFP were expressed from the PEX11 promoter in the atg30Δ strain in methanol medium, and 2 OD cells were collected at indicated time points. Proteins were identified using respective antibodies. SE, short exposure and LE, long exposure.

**Figure 5 cells-11-00157-f005:**
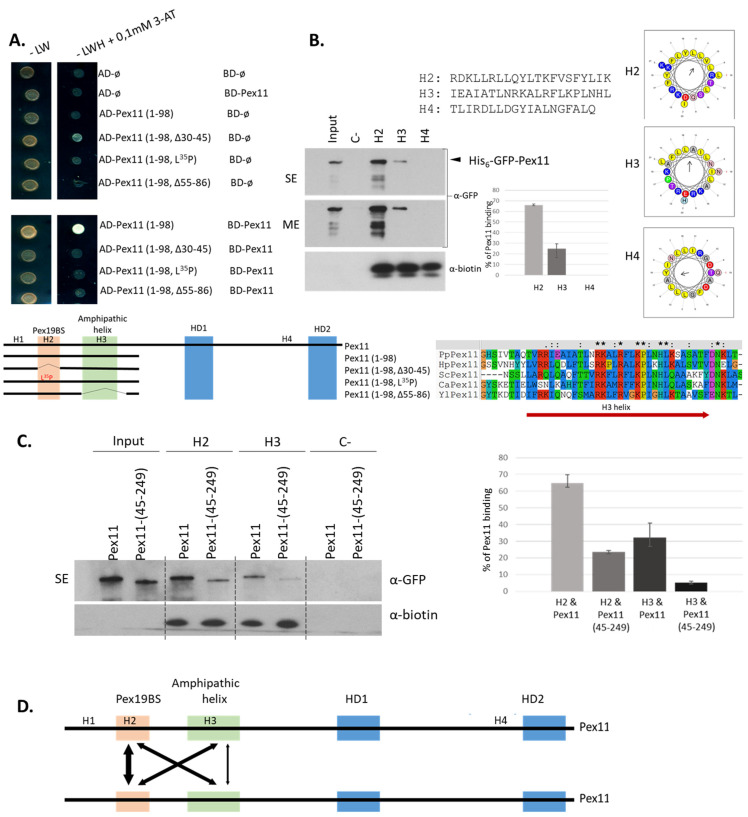
N-terminal amphipathic helices H2 and H3 enable Pex11 dimerization. (**A**). Involvement of N-terminal helices of Pex11 in its self-interaction with Y2H. Full-length Pex11 and its truncated forms, as well as their mutated variants, were fused to either BD or AD domains of GAL4. As negative controls, empty pGBKT7 and pGADT7 were used. Different combinations of these vectors were transformed into the yeast strain Y2H Gold for mapping the regions involved in Pex11 dimerization. Schematic representation of Pex11 truncated and mutant forms used in this study. The H2 helix (aa25-45) containing the Pex19-BS is highlighted in orange, and the H3 (aa55-86) helix, known for lipid binding and dimerization properties, is in green. Sequence alignment of the N-terminal part of Pex11 proteins from various species, showing conservation of specific residues within the H3 helix, whose residues are colored as in Figure 1. Abbreviations and accessions numbers used in sequence alignments: Pp—*Pichia pastoris*, CAY69135; Hp—*Hansenula polymorpha*, DQ645582; Sc—*Saccharomyces cerevisiae*, CAA99168; Ca—*Candida albicans*, EAK92906; Yl—*Yarrowia lipolytica*, CAG81724. 3-AT, 3-amino-1,2,4-triazole; AD, activation domain; BD, DNA binding domain. (**B**). Pex11 preferentially dimerizes via its H2 helix. Pull-down assays using biotinylated peptides corresponding to the H2, H3, or H4 helices (sequences indicated in the figure) bound to streptavidin-coated resin as a bait and His_6_-GFP-Pex11 as a prey. Resins were washed, and proteins were eluted and analyzed by SDS-PAGE. His_6_-GFP-Pex11 was detected by immunoblotting with anti-GFP antibody, and the presence of peptides on the resin was verified by HRP-conjugated streptavidin. Shown is 25% of the input. SE, short exposure; ME, moderate exposure. Quantification of the pull-down assay is shown on the right, with the averages and standard deviations based on three independent sets of experiments. Western blot signals were quantified using the program ImageJ. (**C**). Same as panel B, except full-length His_6_-GFP-Pex11 or truncated form His_6_-GFP-Pex11 (45-249) was used as a prey. Quantification of the Western blots from three independent experiments is shown on the right with standard deviations. (**D**). Graphical representation of homotypic (H2-H2 or H3-H3) and heterotypic (H2-H3) interactions that drive Pex11 dimerization. The thickness of the arrows reflects the strength of the interactions detected by in vitro pull-down assays.

**Figure 6 cells-11-00157-f006:**
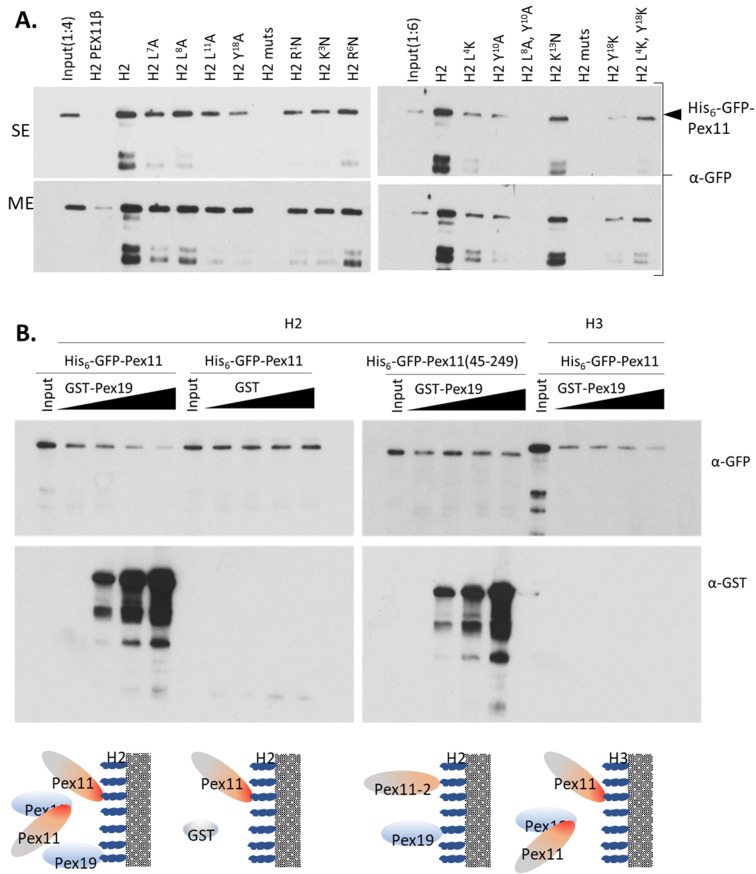
Pex19 binding and Pex11 dimerization to the H2 helix are mutually exclusive. (**A**). Dimerization of Pex11 via its H2 helix involves the same surface as that required for Pex19 binding. Identification of H2 mutants deficient in binding to Pex11 by in vitro pull-down. Biotinylated H2 peptides harboring either the HsPex11β H2 sequence (negative control) or Pex11 H2 WT (positive control) or mutant sequences (see Figure 2 for sequence details) were bound to streptavidin-coated resin prior to the addition of a 4-fold molar excess of His_6_-GFP-Pex11. Levels of captured His_6_-GFP-Pex11 were examined on SDS-PAGE and detected by immunoblotting with anti-GFP antibody. The presence of peptides on the resin was verified by HRP-conjugated streptavidin. Input was diluted 1:4 or 1:6 prior to loading. SE, short exposure; ME, moderate exposure. (**B**). Pex19 competes with Pex11 for the access to H2. Pull-down competition assays of the interaction between the H2 or H3 peptide and His_6_-GFP-Pex11 or His_6_-GFP-Pex11 (45-249) in competition with increasing amounts of the GST-Pex19 or GST alone used as a control are shown. GST-Pex19 or GST alone were added to the resin with bound H2 or H3 simultaneously with His_6_-GFP-Pex11 or His_6_-GFP-Pex11 (45-249). Resins were washed, and proteins were eluted and analyzed by SDS-PAGE. His_6_-GFP-Pex11s, GST-Pex19, and free GST were detected with respective antibodies, and presence of peptides on the resin was verified by HRP-conjugated streptavidin. A graphical illustration of possible interactions for each combination is included.

**Figure 7 cells-11-00157-f007:**
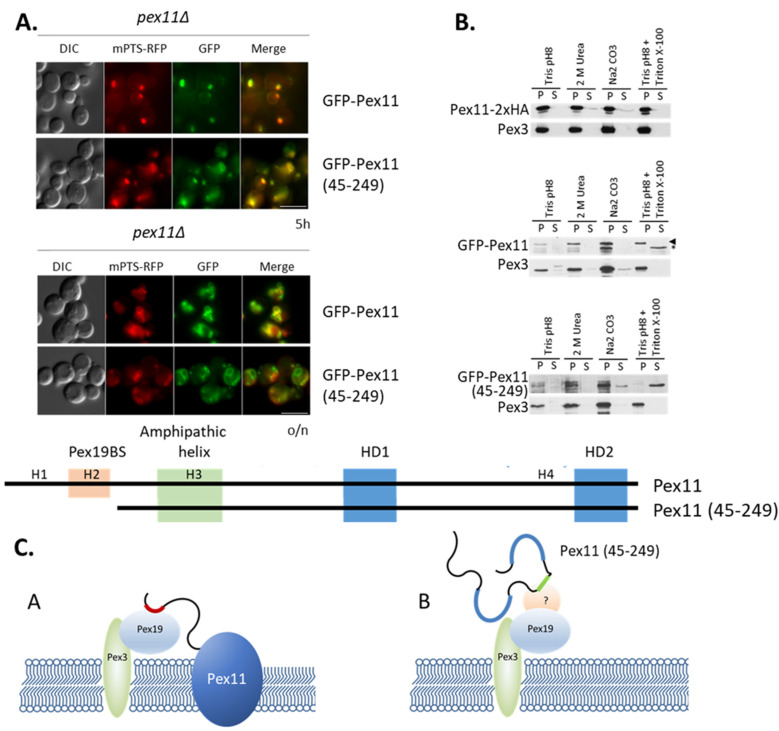
Pex11 import to peroxisomal membranes, but not its trafficking to peroxisomes, requires Pex19 binding. (**A**). Both full-length (GFP-Pex11) and truncated (GFP-Pex11 (45-249)) fusion proteins, respectively, colocalize with the peroxisomal membrane marker, mPTS-RFP. The *pex11*Δ cells containing mPTS-RFP and expressing GFP-tagged Pex11 variants from the endogenous *PEX11* promoter were grown in methanol medium for 5 h (upper panel) or 16 h (o/n-lower panel) for fusion protein induction, prior to observation by fluorescence microscopy. Schematic representation of Pex11 forms used for live-cell imaging is shown below microscopy pictures. Positions of known and predicted modules are highlighted: Pex19-BS (orange), amphipathic helix (green), and hydrophobic regions (HD1 and HD2) predicted to be buried in lipid bilayers (blue). (**B**). Full-length Pex11, but not the Pex11 (45-249) truncated form, is an integral membrane protein. Western blot of membrane protein extraction assay in which the organelle membrane fraction from *pex11*Δ strains expressing Pex11-2HA, GFP-Pex11, and GFP-Pex11 (45-249) were resuspended in four different buffers for peripheral membrane protein extraction (Tris buffer pH 8, 2 mM Urea in Tris buffer pH 8, 0.1 M Na_2_CO_3_ pH 11.5, and Tris buffer pH 8 with Triton X-100) for 30 min at room temperature and fractionated by ultracentrifugation to obtain supernatant (S) and pellet (P) fractions. Proteins were visualized with anti-GFP, anti-HA, and Pex3 antibodies. Pex3 protein was used as a reference integral membrane protein. Note that for GFP-Pex11, the arrowhead indicates proper protein size, and asterisk is probably a truncated form. (**C**). Graphical representation of results. Model (A) presents Pex11 trafficking and incorporation into peroxisomal membranes with the assistance of Pex19 (and Pex3). Model (B) presents Pex11 (45-249) trafficking to, but not its insertion, into the peroxisome membrane, due to the lack of Pex19-BS. It is unclear if Pex11 (45-249) requires interaction with another protein (marked as “?”) for its targeting to peroxisomes. Bar = 5 μm.

**Figure 8 cells-11-00157-f008:**
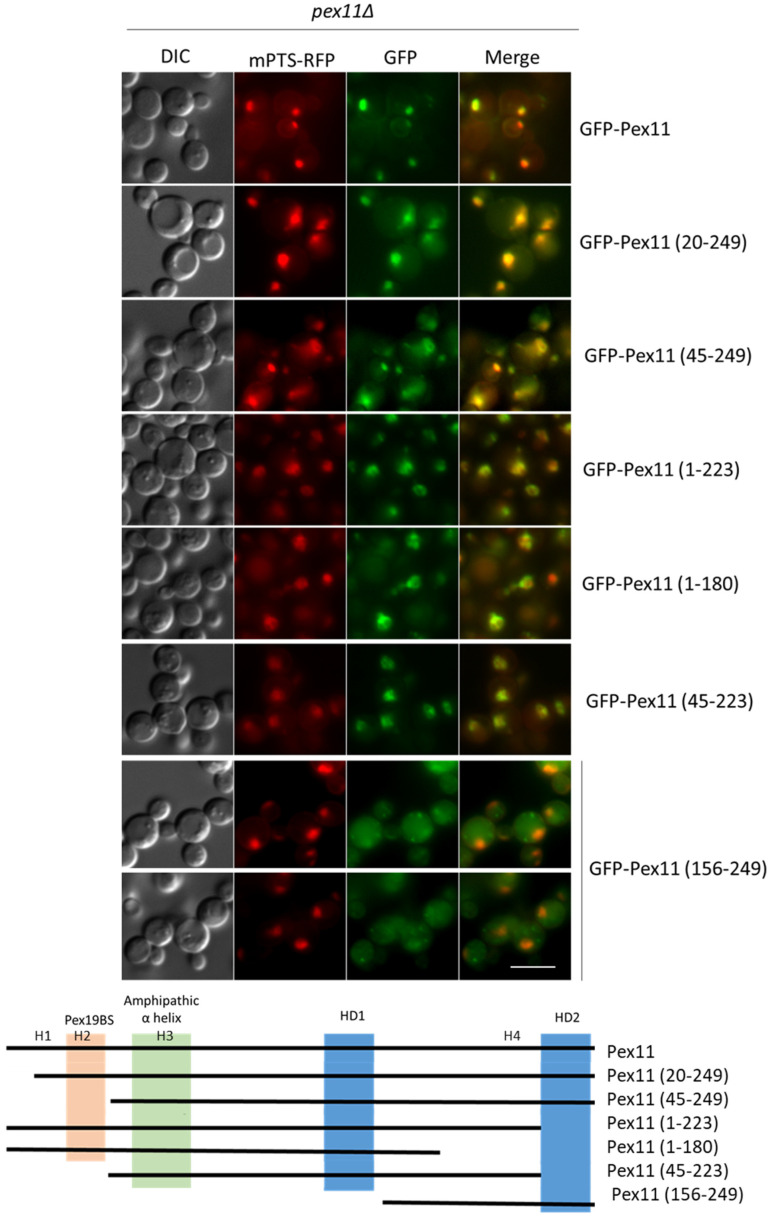
Pex11 has a Pex19-independent mPTS that is distinct from the Pex19-BS. Fluorescence microscopy images of *pex11*Δ cells expressing full-length and truncated GFP-Pex11 fusion proteins and mPTS-RFP for peroxisome visualization after 5 h in methanol medium. Bar = 5μm. A schematic representation of Pex11 truncated forms used in this study is shown below the micrographs. Positions of known and predicted modules are highlighted as shown in Figure 5: Pex19-BS (orange); amphipathic helix (green); and hydrophobic regions, HD1, and HD2, predicted to be buried in lipid bilayers (blue).

**Figure 9 cells-11-00157-f009:**
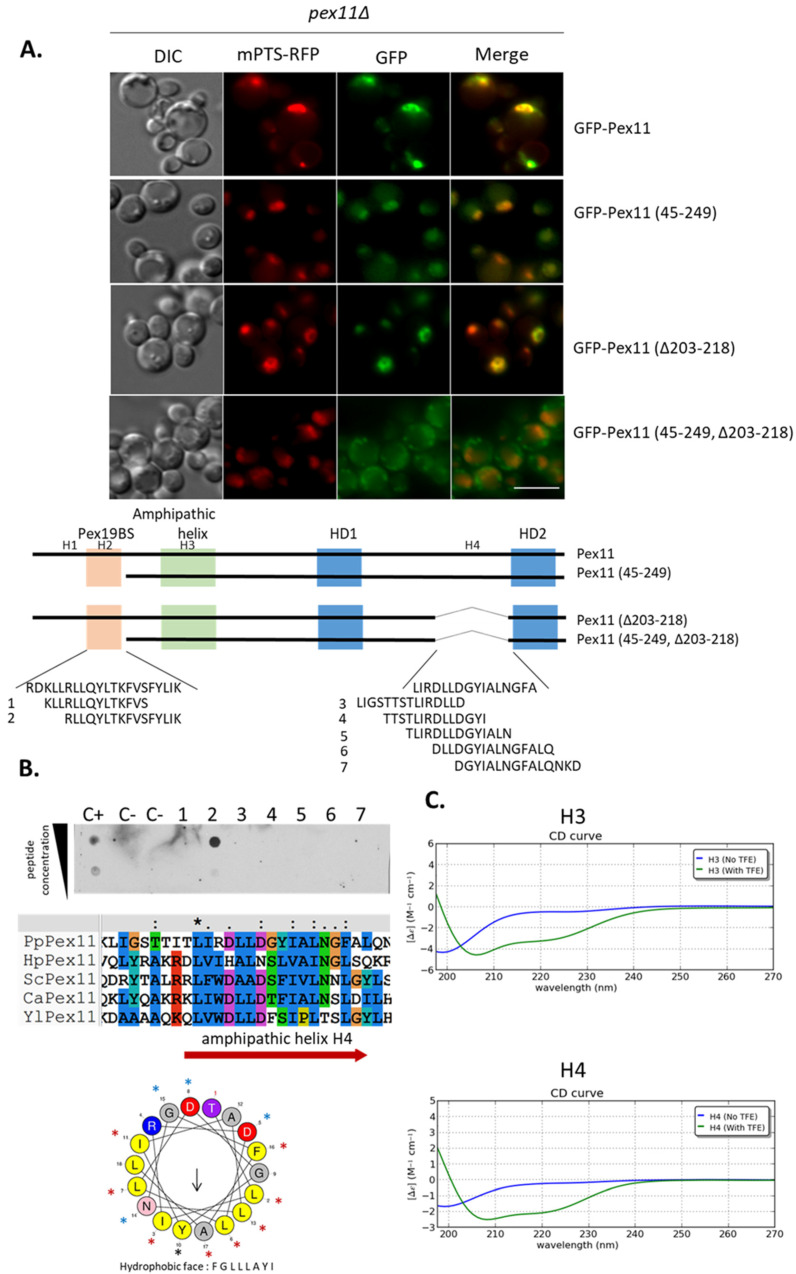
Amphipathic helix H4 of Pex11 has a Pex19-independent mPTS. (**A**). Fluorescence microscopy images of *pex11*Δ cells expressing full-length and truncated GFP-Pex11 fusion proteins and mPTS-RFP for peroxisome visualization after 16 h in methanol medium. A schematic representation of Pex11 truncated forms used in this study is shown below the micrographs. Positions of known and predicted modules are highlighted similarly as in Figure 5. Bar = 5 μm. (**B**). The H4 (aa203-218) region in Pex11 is not recognized by Pex19. Short 14 and 15-mer peptides with three amino acid shifts spanning the identified mPTS with its flanking N- and C-terminal residues were synthesized and subjected to the Pex19 in vitro binding assay. Dot-blot analysis was done as described in Figure 2B. Peptides and proper controls (including peptides described in Figure 2B) were spotted on a nitrocellulose membrane at two concentrations (20 nmol and 5 nmol) and tested for interaction with GST-Pex19. Bound protein was detected immunologically with polyclonal anti-Pex19 antibodies. Sequence alignment of regions from various Pex11s corresponding to identified mPTS in Pex11 is shown at the bottom. Residues were colored by Clustal X [38], based on their physico-chemical properties as in Figure 2B. The position of the amphipathic helix H4 in Pex11 is marked by a red arrow, and its helical wheel plot generated using HeliQuest is shown on the right. The black arrow in the plot points to the hydrophobic face, and its length corresponds to the hydrophobic moment. (**C**). The secondary structures of the H3 (used as reference) and H4 peptides were analysed by CD spectroscopy. The spectrum shows that, like the H3 peptide, H4 is unstructured in phosphate buffer, but α-helical in 30% TFE. Abbreviations and accession numbers used in sequence alignments: Pp—*Pichia pastoris*, CAY69135; Hp—*Hansenula polymorpha*, DQ645582; Sc—*Saccharomyces cerevisae*, CAA99168; Ca—*Candida albicans*, EAK92906; Yl—*Yarrowia lipolytica*, CAG81724.

## Data Availability

All study data are included in the article and/or Appendix A.

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
