# Peer review of "Recognition and Chaperoning by Pex19, Followed by Trafficking and Membrane Insertion of the Peroxisome Proliferation Protein, Pex11"

_cells, 2022, doi:10.3390/cells11010157_

Round 1
Reviewer 1 Report
The manuscript of Zientara-Rytter et al. (entitled Recognition and chaperoning by Pex19, followed by trafficking and membrane insertion of the peroxisome proliferation protein, Pex11) explores the dual function of Pex19 protein (i.e. transporter and chaperone) and its impact on the proper peroxisomal trafficking of Pex11 in yeast (Pichia pastoris/Komagataella phaffi). The paper elucidates an important point in molecular mechanisms involved in peroxisomal biogenesis and proliferation.
The subject of the work would be of interest for the scientific community, and it fits to the previous works of this group in the field.
The experimental design is correct, and the results are clearly presented. The conclusions are convincing and supported by the data.
Major point: none
Minor point:
- The abbreviation mPTS is not defined in the abstract (Page 1, Line 16).
Author Response
Minor issue has been addressed.
Reviewer 2 Report
cells Ms. ID: cells-1500612
Title: Recognition and chaperoning by Pex19, followed by trafficking and membrane insertion of the peroxisome proliferation protein, Pex11
Authors: Katarzyna et al.
Remarks to the Authors:
The authors performed a comprehensive analysis of how the Pichia pastoris Pex11 is recognized and chaperoned by Pex19, targeted to peroxisome membranes and inserted therein. Pex11 contains one Pex19-binding site (Pex19-BS) that is required for insertion of Pex11 into peroxisome membranes. The authors also found that Pex11 has a separate Pex19-independnet mPTS. Furthermore, the authors showed that Pex19 acts as a chaperone by interacting with the Pex19-BS in Pex11, thereby protecting Pex11 from spontaneous oligomerization of Pex11. From these studies, the authors proposed that Pex19 is required for insertion of Pex11 into peroxisomal membrane, but not essential for its trafficking to peroxisome surface. However, some issues remain to be addressed to draw the conclusion.
Major concerns:
- The authors describe that Pex19-BS of Pex11 serves as mPTS as shown by the peroxisomal localization of GFP-Pex11(Δ203-218) in Figure 9A. However, the quality of the data is not enough to agree with the peroxisomal localization.
- The authors describe that aa203-218 of Pex11 acts as mPTS from the findings that the lack of aa203-218 interfered with the peroxisomal targeting of GFP-Pex11 (45-249). However, it is not clear if H4 region serves as a mPTS at physiological condition. This should be addressed.
- On page10: The reduced Pex19 binding ability by the substitution of Tyr (aa10) and Leu (aa7) is not clear.
- The signal of H4 peptide was not detected in Figure 5C. Why?
- Endogenous peroxisomal peripheral membrane protein should be included as a positive control in Figure 7B.
Minor concerns:
- The reason why the authors used Pex11 (Δ84-134) for generating the deletion mutant is required.
- “homo-oligomerization” and “dimerization” were likely used confusingly in the text.
- On page 4, line 149: Rabbit anti-ScF1β might not be used in this study.
- On page 10, line 350: Is there any data showing that GST does not bind to any of the peptides used in Figure 3A?
- On page 23, lines 710-711: It is unclear what the authors mean by the statement.
Author Response
We thank the reviewer for the comments and questions.
Major concerns:
- The authors describe that Pex19-BS of Pex11 serves as mPTS as shown by the peroxisomal localization of GFP-Pex11(Δ203-218) in Figure 9A. However, the quality of the data is not enough to agree with the peroxisomal localization.
A new Fig. 9 is included.
- The authors describe that aa203-218 of Pex11 acts as mPTS from the findings that the lack of aa203-218 interfered with the peroxisomal targeting of GFP-Pex11 (45-249). However, it is not clear if H4 region serves as a mPTS at physiological condition. This should be addressed.
Testing the physiological relevance of the Pex19-independent mPTS in H4 is technically impossible at present, because in the presence of a Pex19-dependent mPTS in the H2 helix of Pex11, the only way to truly address the physiological relevance of the Pex19-independent mPTS in helix H4 would be to test a H2 mutant that is deficient in peroxisomal targeting (i.e. dysfunctional Pex19-dependent mPTS), but that still binds Pex19 (needed for membrane insertion of the mutant Pex11). Unfortunately, so far we have been unable to separate the peroxisomal targeting and Pex19-binding roles in the H2 region.
On page10: The reduced Pex19 binding ability by the substitution of Tyr (aa10) and Leu (aa7) is not clear.
The reviewer may have looked at the Ponceau S staining, showing equal loading of peptides, rather than at the immunoblots below showing reduced binding of these Pex11 mutants to Pex19 (Fig. 3A).
The signal of H4 peptide was not detected in Figure 5C. Why?
This is expected if H4 has no role in Pex11 oligomerization.
Endogenous peroxisomal peripheral membrane protein should be included as a positive control in Figure 7B.
We do not believe this is essential. The purpose of this study was to determine if full length Pex11 and its truncated form Pex11 (45-249) are integral membrane proteins. Thus, as the appropriate control we included another integral PMP, Pex3, as a marker.
Minor concerns:
- The reason why the authors used Pex11 (Δ84-134) for generating the deletion mutant is required.
Clarified.
- “homo-oligomerization” and “dimerization” were likely used confusingly in the text.
Corrected.
- On page 4, line 149: Rabbit anti-ScF1β might not be used in this study.
Our custom-made rabbit anti-ScF1 β antibody recognizes also F1β from Pichia pastoris.
- On page 10, line 350: Is there any data showing that GST does not bind to any of the peptides used in Figure 3A?
All our peptides were analyzed for binding to GST-PpPex19 and to GST alone. Since there was no binding of free GST to any of tested variants of H2 peptides, we did not include this data for the clarity of the figure, but state the result as not shown.
- On page 23, lines 710-711: It is unclear what the authors mean by the statement.
The sub-title has been modified for clarity. It now reads “Distinct roles of Pex19 in chaperoning and membrane insertion of Pex11, and in peroxisome targeting”
Round 2
Reviewer 2 Report
Remarks to the Authors:
The authors responded well to the reviewer’s comments. However, this reviewer has still one concern about the data presented in Fig. 5C.
This reviewer expected that H4 peptide signal is supposed to be detected by HRP-conjugated streptavidin in Fig. 5C as in Fig. 5B. But the result was not the case. Is it expected that H4 peptide was not detected in Fig. 5C?
Author Response
We thank the reviewer for flagging this mislabeling. We have fixed this by removing the blank lanes corresponding to the H4 peptide in Fig. 5C. Importantly, the results are unchanged, especially as shown in Fig. S5, where the biotin signal for peptide H4 is present, but it still does not bind either full-length Pex11-GFP, Pex11(45-249)-GFP or Pex11(Δ203-218)GFP. We have made related changes in the text and Fig. 5 legend (see Track changes).